# Adversarial Test Case Generation via Reinforcement Learning Extends Scaling Laws

## Abstract

Rule-based reinforcement learning (RL) has greatly advanced the coding capabilities of large language models (LLMs). However, existing RL methods remain largely confined to code generation, relying on fixed test cases for evaluation and leaving the problem of test case generation underexplored. Generating diverse and adversarial test cases is critical, as it not only enriches coding knowledge but also enables effective self-verification during inference. Recent supervised learning approaches attempt to jointly train code and test case generation during the post-training stage. Yet, these methods fall short: supervised learning inherently lags behind RL in coding performance, and the resulting test cases often lack diversity and adversarial quality, limiting their ability to identify erroneous code. To address these limitations while retaining the advantages of RL, we propose Test Cases Scaling (TCS), a two-stage reinforcement learning framework for learning to generate high-quality adversarial test cases. TCS employs stage-specific reward functions and a policy-aligned training buffer to progressively enhance test case quality and alignment with the evolving model. Experimental results on TACO and LiveCodeBench show that TCS significantly outperforms supervised baselines in both code and test case generation during both training and inference. Furthermore, adversarial test cases generated by our trained TCS-7B model improve the inference-time performance of leading proprietary LLMs.

## 1 Introduction

Code generation has emerged as a compelling application of large language models (LLMs), enhancing the productivity of professional developers while simultaneously lowering the barrier for non-experts through coding assistants (Dong et al., 2025). The field has progressed from early heuristic systems and expert-rule frameworks to modern approaches built upon large-scale, fine-tuned code LLMs (Gulwani, 2010; Roziere et al., 2023; Hui et al., 2024a; Dong et al., 2025). Among these, reinforcement learning (RL) has proven particularly effective: code correctness can be automatically verified through predefined test cases, enabling RL to surpass traditional supervised fine-tuning (SFT), which relies on costly human-curated code generation data (Shojaee et al., 2023; Le et al., 2022; Luo et al., 2025).

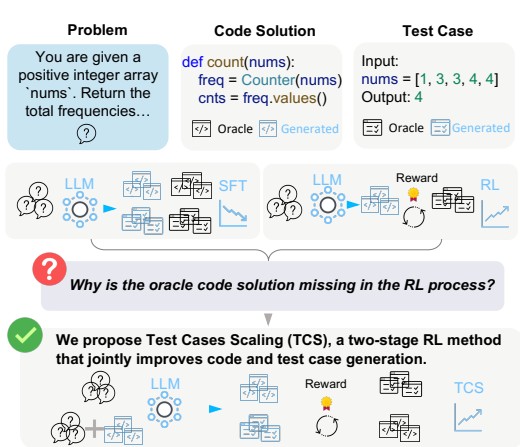

Figure 1: TCS enables adversarial test case generation using oracle code solutions.

Recent advancements in code generation have led to studies on models' ability to generate test cases for self-verification during inference. One approach uses prompt engineering to directly elicit test cases from the model (Chen et al., 2023; Li & Yuan, 2024), while another employs supervised learning to build datasets for test case generation (Lin et al., 2025; Zhang et al., 2024; Jiang et al., 2024a; Zhong et al., 2024). However, the former is limited by model capacity, and the latter by static datasets that restrict diversity and adversarial quality. RL has largely ignored this task due to

the difficulty in defining effective reward functions. In code generation, correctness can be easily verified by passing a predefined test suite, but the value of a generated test case is less clear. Even if the outputs align with the oracle code solution, the test case may still be trivial. RL's sensitivity to reward functions means lenient rewards risk reward hacking, while strict rewards may impede feedback (Chen et al., 2024; Fu et al., 2025).

Despite these challenges, test case generation is crucial in RL for enhancing models' comprehensive coding capabilities (Kaplan et al., 2020; Hoffmann et al., 2022). Additionally, generated test cases enable inference-time scaling beyond reward-model supervision, acting as task-specific tools for coding (Zhong et al., 2024; Zhang et al., 2024; Lin et al., 2025). To train LLMs to generate high-quality test cases through RL, we propose a unified framework that simultaneously trains the model to generate both code and test cases. Unlike typical RL methods for code generation that use problem–test case pairs, our method (fig. 1) uses triples <problem, solution, test case>. We train code generation with the problem and test case, and test case generation with the problem and solution. Since generating test cases is not a primary task in open-source base models, enabling them to produce accurate and challenging test cases presents significant challenges (Roziere et al., 2023; Lin et al., 2025). We propose a two-stage reinforcement learning approach, **Test Cases Scaling (TCS)**, equipped with tailored reward functions to guide each stage effectively. The first stage trains the LLM to generate correct test cases, and the second stage trains it to generate adversarial test cases that identify errors in incorrect solutions. This enhanced ability allows us to sample multiple candidate test cases, execute each candidate code solution, and select the code with the highest empirical pass rate at inference time.

Through comprehensive evaluations on the TACO and LiveCodeBench benchmarks, we demonstrate that our method consistently outperforms baselines in both training-time and inference-time performance. Moreover, our TCS-7B model can also benefit leading proprietary models by improving their inference-time scaling ability. Our contributions are threefold:

- We identify a critical gap in current RL frameworks for coding abilities: the neglected role of test case generation and the resulting deficiency in models' abilities to generate effective test cases.
- We propose a novel adversarial reward design specifically for test case generation, implemented through a two-stage reinforcement learning approach that effectively improves the models' ability to identify errors in code.
- Our experiments demonstrate the significant advantages of applying RL to test case generation, and the resulting TCS-7B model can effectively guide leading proprietary models.

## 2 BACKGROUND

### 2.1 REINFORCEMENT LEARNING

Reinforcement learning (RL) effectively enhances LLMs beyond supervised fine-tuning by optimizing specific objectives via reward signals. In LLMs, RL adapts a pre-trained model $\pi_\theta$ to maximize a task-specific reward function. For a prompt $x \sim \mathcal{D}$, the model generates a completion $y \sim \pi_\theta(\cdot \mid x)$ and receives scalar feedback $r(x, y)$, typically in the range $[0, 1]$. The standard optimization objective is: x

$$J(\theta) = \mathbb{E}_{x \sim \mathcal{D}} \, \mathbb{E}_{y \sim \pi_\theta(\cdot|x)} \big[ r(x, y) \big]. \tag{1}$$

Proximal Policy Optimization (PPO) is widely used for LLM training but requires a separate value function model, adding computational overhead. We use Group Relative Policy Optimization (GRPO) (Shao et al., 2024) to avoid this by using the average reward of multiple outputs sampled for the same prompt as a baseline. GRPO samples a group of $G$ outputs $\{y_i\}_{i=1}^G$ for each prompt $x$, with advantages calculated relative to the average reward within the group:

$$\mathcal{J}(\theta) = \mathbb{E}_{x \in \mathcal{D}, \{y_i\}_{i=1}^G \sim \pi_{\theta_{\text{old}}}(\cdot|x)} \left[ \frac{1}{G} \sum_{i=1}^G \left( \min \left( \rho_i A_i, \text{clip} \left( \rho_i, 1 - \epsilon, 1 + \epsilon \right) A_i \right) \right) - \beta \mathbb{D}_{\text{KL}}(\pi_\theta \| \pi_{\text{ref}}) \right], \tag{2}$$

where $\rho_i = \frac{\pi_\theta(y_i|x)}{\pi_{\theta_{\text{old}}}(y_i|x)}$ denotes the importance sampling ratio, $\pi_{\text{old}}$ is the old policy parameters for sampling, $\epsilon$ and $\beta$ are two hyper-parameters, and $\mathbb{D}_{\text{KL}}(\cdot \| \cdot)$ is the KL-divergence. The advantage $A_i$ of the generation $y_i$ is defined as

$$A_i = \frac{r_i - \text{mean}(r_1, \ldots, r_G)}{\text{std}(r_1, \ldots, r_G)}, \tag{3}$$

where $r_i$ is the reward on response $y_i$.

## 2.2 CODE AND TEST CASE GENERATION

The task of code generation involves creating a program $C$ from a natural-language problem description $P$ that fulfills all specified requirements. Due to the lack of a standard implementation for code solutions, practitioners must construct test suites $\mathcal{T} = \{(I_i, O_i)\}_{i=1}^m$ to verify if a program $C$ meets the problem requirements. A candidate program $C$ is deemed correct if, for every $i$,

$$\text{Exec}(C, I_i) = O_i. \tag{4}$$

In RL for code generation, a standard reward function $R^c$ is defined to evaluate the correctness of a generated program $C$ against an assumed oracle test suite $\mathcal{T}^*$ (Yu et al., 2025):

$$R^c(C, \mathcal{T}^*) = \begin{cases} 1 & \text{if } \forall(I_i, O_i) \in \mathcal{T}^* : \text{Exec}(C, I_i) = O_i \\ 0 & \text{otherwise} \end{cases} \tag{5}$$

Thus, test case coverage is essential. A valid test suite must meet two criteria: for every correct program, each input should yield the expected output; for every incorrect program, at least one test case should produce an output that deviates from the expected result. Current code generation research mainly targets the creation of correct programs, presuming the validity of test suites.

Building a comprehensive test suite is inherently challenging. In creating programming problems, humans often provide both correct and incorrect solutions, manually design critical test cases, and algorithmically generate random cases to enhance coverage. Despite these efforts, ensuring the detection of all flawed implementations remains difficult. Consequently, some algorithmic competitions allow extra time post-coding for participants to review other contestants' solutions that passed initial tests, encouraging them to identify test cases that could cause these solutions to fail. This underscores that skilled competitive programmers are proficient in constructing effective test cases.

# 3 TEST CASES SCALING (TCS)

## 3.1 PROBLEM FORMULATION

Each problem in our prepared dataset $\mathcal{D}$ contains a problem description $P$, a ground-truth solution $C^*$, and a test suite $\mathcal{T} = \{(I_i, O_i)\}_{i=1}^m$. This setup allows us to define two distinct roles for our LLM:

- **Solver**: Given a problem description $P$, the solver's objective is to generate code $C$ that passes all test cases in the test suite $\mathcal{T}$, i.e., $\forall(I_i, O_i) \in \mathcal{T} : \text{Exec}(C, I_i) = O_i$.
- **Verifier**: Given a problem description $P$ and generated code $C$, the verifier aims to produce an effective test case $(I_v, O_v)$ that can validate the correctness of the given code. Specifically, if $C$ is correct, then $\text{Exec}(C, I_v) = O_v$; if $C$ is incorrect, then $\text{Exec}(C, I_v) \neq O_v$ while $\text{Exec}(C^*, I_v) = O_v$.

Unlike existing RL approaches that focus solely on training LLMs to be effective **Solvers**, our method aims to develop LLMs that can also serve as proficient **Verifiers**, further leveraging information from ground truth solutions. We seek to enable the model to emulate expert programmers who not only solve problems but also accurately identify vulnerabilities in incorrect code. This dual capability enhances both code generation skills and the ability to assess code correctness, resulting in scaling law benefits during both training and inference. Specifically, the ability to write effective test cases not only strengthens code generation but also allows the model to efficiently select correct solutions from multiple generated candidates during inference.

## 3.2 RL TRAINING

### 3.2.1 REWARD FUNCTION

Reward design is the cornerstone of reinforcement learning, directly determining training outcomes. Unlike mathematical problems with definitive solutions or code generation with assumed correct test suites, defining an effective reward function for test case generation presents unique challenges. Overly lenient designs can lead to reward hacking, while excessively strict requirements make it difficult for LLMs to earn rewards.

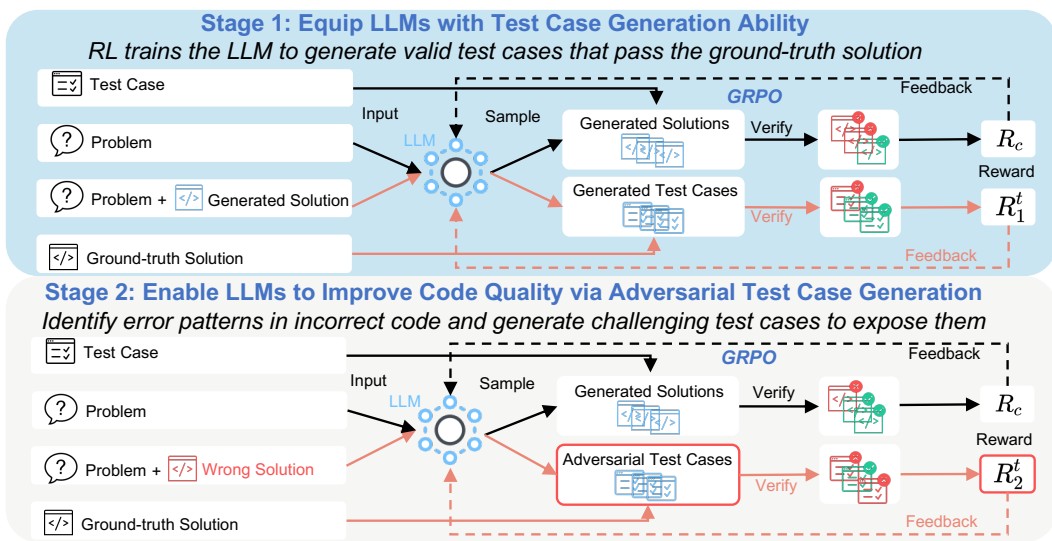

Figure 2: Overview of the two-stage training process in Test Cases Scaling (TCS).

For a ground-truth solution $C^*$ and a generated test case $(I_g, O_g)$, our initial approach was straight-forward: if the actual output from the ground-truth solution $\text{Exec}(C^*, I_g)$ matches the expected output $O_g$, we assign a reward of 1; otherwise, 0. However, during training, models ignored prompt instructions and engaged in reward hacking by copying example test cases. To counter this, we imposed a constraint that generated test cases must differ from examples. The reward function $R_1^t$ is defined as:

$$R_1^t(I_g, O_g) = \begin{cases} 1 & \text{if } \text{Exec}(C^*, I_g) = O_g \text{ and } (I_g, O_g) \notin \mathcal{T}_{\text{example}} \\ 0 & \text{otherwise} \end{cases} \tag{6}$$

where $\mathcal{T}_{\text{example}}$ denotes the example test cases in the problem description.

While $R_1^t$ enables the model to generate valid test cases, these cases often lack effectiveness in identifying erroneous code. To maximize rewards, the LLM tends to output simple test cases that easily pass verification against the ground-truth solution. We therefore introduced an adversarial approach to generate test cases that not only pass the ground-truth solution but also expose vulnerabilities in incorrect code. We define $R_2^t$ as:

$$R_2^t(I_g, O_g, C^*, C_{\text{wrong}}) = \begin{cases} 1 & \text{if } \text{Exec}(C^*, I_g) = O_g \text{ and } \text{Exec}(C_{\text{wrong}}, I_g) \neq O_g \\ & \quad \text{and } (I_g, O_g) \notin \mathcal{T}_{\text{example}} \\ 0 & \text{otherwise} \end{cases} \tag{7}$$

Despite $R_2^t$'s theoretical soundness, we found it challenging to obtain effective reward signals during initial training, especially for models lacking test case generation ability. For models like DeepSeek-R1-Distill-Qwen-1.5B, training with $R_2^t$ yields almost no reward signals for extended periods. This occurs because models without prior test case generation capability struggle to immediately produce challenging test cases that uncover subtle code errors. We therefore propose a two-stage reinforcement learning approach: first using $R_1^t$ to develop basic test case generation ability, then transitioning to $R_2^t$ to develop adversarial test case generation skills.

### 3.2.2 Two-Stage Reinforcement Learning

Our method jointly trains both code generation and test case generation capabilities, as shown in Figure 2. At each training step, we sample a mixture of data from both datasets and apply the corresponding reward functions: $R^c$ for code generation and either $R_1^t$ or $R_2^t$ for test case generation.

**Policy-aligned buffer.** To ensure diversity and foster the generation of adversarial test cases, we construct test case generation prompts dynamically using code produced in real time during training, rather than relying on a fixed dataset. Specifically, code outputs from the code generation tasks that satisfy predefined criteria are collected into a dynamic policy-aligned buffer $\mathcal{B}$, which serves as the

---

**Algorithm 1** Test Case Scaling

---

1: **Input:** Dataset $\mathcal{D}$, initial policy $\pi_\theta$, buffer size $T_b$, group size $G$, total training steps $T$
2: **Initialize:** Policy-aligned buffer $\mathcal{B} \leftarrow \emptyset$
3: **for** $t = 1, \ldots, T$ **do**
4:     Sample a data batch from the joint dataset $\mathcal{D} \cup \mathcal{B}$
5:     **for** input prompt $x$ in the batch **do**
6:         Generate a group of responses $\{y_i\}_{i=1}^{G}$ from $\pi_\theta$
7:         **if** the input $x$ is from $\mathcal{D}$ **then**
8:             ▷ Code generation problem
9:             Compute rewards $r^c$ for each generation $y_1, \ldots, y_G$ according to eq. (5)
10:          $\mathcal{B} \leftarrow \mathcal{B} \cup \{(x, y_i) \mid i = 1, \ldots, G, \ (x, y_i)$ satisfies the buffer policy$\}$
11:         **else**
12:             ▷ Test case generation problem
13:             Compute reward $r_1^t$ or $r_2^t$ for each generation $y_1, \ldots, y_G$ according to eq. (6) or eq. (7)
14:             Remove $x$ from $\mathcal{B}$
15:         **end if**
16:     **end for**
17:     Update the model $\pi_\theta$ according to eq. (2)
18:     Remove data samples from $\mathcal{B}$ that were collected before $t - T_b$ steps
19: **end for**

---

reservoir for constructing training instances of test case generation. To further maintain diversity and ensure alignment with the current capabilities of the model, previously sampled instances are discarded, and only samples from the most recent $T_b$ training steps are retained. The prompt template for constructing test case generation tasks is given as follows:

---

**Prompt Template for Adversarial Test Case Generation**

```
You are an expert TEST CASE GENERATOR for programming competitions.
...
You have been provided with:
* PROBLEM DESCRIPTION: {problem}
* A SOLUTION CODE THAT MAY CONTAIN LOGIC ERRORS: {code}
Your goal is to create a test case that is valid under the problem
constraints and is likely to expose incorrect behavior in the
↪  provided
code if such errors exist.
...
```

---

**Stage 1.** The first stage aims to develop the model's ability to generate valid and diverse test cases. We collect code samples from the training process that produce successful outputs, excluding those with timeouts or syntax errors, and add them to the policy-aligned buffer $\mathcal{B}$ for test case generation prompts. All tasks in this stage use $R_1^t$ as the reward function, focusing on generating correct test cases that pass the ground truth solution, with dynamic code sampling for prompt construction ensuring diversity.

**Stage 2.** Upon reaching a predefined test case generation accuracy in stage one, the process advances to stage two. Here, only incorrect code from code generation training is retained in $\mathcal{B}$. Subsequent test case generation tasks are evaluated with $R_2^t$. This stage aims to train the model to recognize error patterns in diverse erroneous implementations and generate adversarial test cases to expose these errors.

We optimize LLM parameters for both code and test case generation using the GRPO algorithm, leveraging collected inputs, outputs, and rewards. The full procedure is detailed in algorithm 1.

### 3.3 INFERENCE-TIME SCALING

Conventional inference-time scaling methods typically involve sampling $N$ candidate responses simultaneously and then using a reward model to select the optimal final answer. Our approach provides an alternative form of inference-time scaling specifically tailored for code generation. For

Table 1: Performance on TACO and LiveCodeBench

| Model | TACO | | LiveCodeBench | |
|---|---|---|---|---|
| | w/o pub | w/ pub | w/o pub | w/ pub |
| **DeepSeek-R1-Distill-Qwen-1.5B** | | | | |
| Base Model | 5.63 | | 14.38 | |
|   + Reward Model | 12.70 | 14.60 | 23.30 | 30.11 |
|   + Self-Generated Test Cases | 5.78 | 13.91 | 22.00 | 30.09 |
| SFT Model | 9.43 | | 17.21 | |
|   + Reward Model | 14.32 | 16.78 | 23.67 | 32.45 |
|   + Self-Generated Test Cases | 15.12 | 16.23 | 24.33 | 31.89 |
| **RL using TCS Training** | 12.31 | | 20.63 | |
|   + Reward Model | 15.66 | 23.17 | 24.01 | 37.99 |
|   **+ Self-Generated Test Cases** | **20.52** | **25.18** | **27.47** | **38.90** |
| **DeepSeek-R1-Distill-Qwen-7B** | | | | |
| Base Model | 14.36 | | 28.56 | |
|   + Reward Model | 24.87 | 26.98 | 42.65 | 46.24 |
|   + Self-Generated Test Cases | 18.67 | 26.02 | 43.01 | 46.15 |
| SFT Model | 18.92 | | 32.14 | |
|   + Reward Model | 27.45 | 29.67 | 43.23 | 48.91 |
|   + Self-Generated Test Cases | 27.32 | 29.78 | 44.67 | 47.82 |
| **RL using TCS Training** | 24.09 | | 37.03 | |
|   + Reward Model | 31.11 | 37.67 | 43.01 | 53.40 |
|   **+ Self-Generated Test Cases** | **35.35** | **39.40** | **48.79** | **54.75** |

each problem, we construct a test case generation prompt using the same methodology as in the training phase, encouraging the model to identify potential errors in the code. The LLM then generates $M$ test cases for this prompt, resulting in a total of $N \times M$ test cases.

We execute all $N$ candidate responses against these $N \times M$ test cases and select the candidate that passes the most test cases as the final answer. This approach leverages the model's own test case generation capabilities to evaluate the robustness of its code solutions. Formally, our selection criterion can be expressed as:

$$C_{\text{selected}} = \underset{C_i \in \{C_1, C_2, ..., C_N\}}{\arg\max} \sum_{j=1}^{N \times M} \mathbb{I}[\text{Exec}(C_i, I_j) = O_j] \qquad (8)$$

where $C_i$ represents the $i$-th candidate code solution, $I_j$ and $O_j$ are the input and expected output of the $j$-th test case, and $\mathbb{I}$ is the indicator function that equals 1 when the execution matches the expected output and 0 otherwise.

# 4 EXPERIMENTS

## 4.1 EXPERIMENTAL SETUP

Our method requires high-quality data comprising a problem description, a ground-truth solution, and a reliable suite of test cases. We use the TACO (Li et al., 2023) dataset, containing 25,433 problems from coding platforms, with some providing ground-truth solutions. To ensure quality, we filter for problems with enough test cases and retain those where at least one Python solution passes all test cases, resulting in a curated set of 6,318 problems. For evaluation, we use the TACO validation set of 1,000 problems and LiveCodeBench (Jain et al., 2024), a benchmark of recent competitive programming problems. Training and evaluation details are in Appendix A.

## 4.2 MAIN RESULTS

We assess the DeepSeek-R1-Distill-Qwen models with 1.5B and 7B parameters in three scenarios: (1) base models without further training, (2) base models fine-tuned with SFT on joint code–test case generation tasks, and (3) our TCS method that jointly trains on code and test case generation.

For the SFT dataset, we used the DeepSeek-R1-Distill-Qwen 32B model for sample generation and filtering following Sol-Ver (Lin et al., 2025). In TCS, we proceed to the second stage once test case generation accuracy surpasses a preset threshold. All methods use a maximum response length of 8192 tokens to balance performance and efficiency during training and inference.

Results are shown in three rows per setting. The first row displays pass@1 scores (averaged over 16 samples for TACO and 32 for LiveCodeBench). The second and third rows show Best-of-N (BoN) scores for two inference-time scaling strategies: reward model selection and self-generated test case selection, with N=16 for TACO and N=32 for LiveCodeBench. We use InternLM2-7B-reward (Cai et al., 2024), a state-of-the-art reward model at the same scale, for reward model selection. We adopt the CodeT (Chen et al., 2023) pipeline approach to select code solutions using self-generated test cases. Performance is reported with and without public test cases, which are optional in coding tasks.

Table 1 provides a comprehensive comparison of training-time and inference-time performance. TCS yields substantial pass@1 improvements for the 1.5B and 7B models compared to both the base and SFT models, aligning with current consensus (Trung et al., 2024). In inference-time settings, TCS fine-tuned models with self-generated test case selection consistently show significant gains. For the base model, self-generated test case selection performs notably worse than reward model selection, indicating limited inherent ability to generate high-quality test cases. The SFT model shows moderate improvements after training on test case generation, yet when candidate solutions are filtered with public test cases, it fails to outperform the reward model. When comparing inference-time scaling methods for TCS models, self-generated test cases generally outperform reward model selection, especially without public test cases to filter candidates. This suggests that the self-generated method is more robust to the number of candidate solutions than the reward model. A plausible explanation is that the reward model is more sensitive to out-of-distribution data, with larger candidate pools increasing the likelihood of encountering such data. We provide a deeper analysis of this phenomenon in Section 4.4.

### 4.3 TEST CASE EFFECTIVENESS

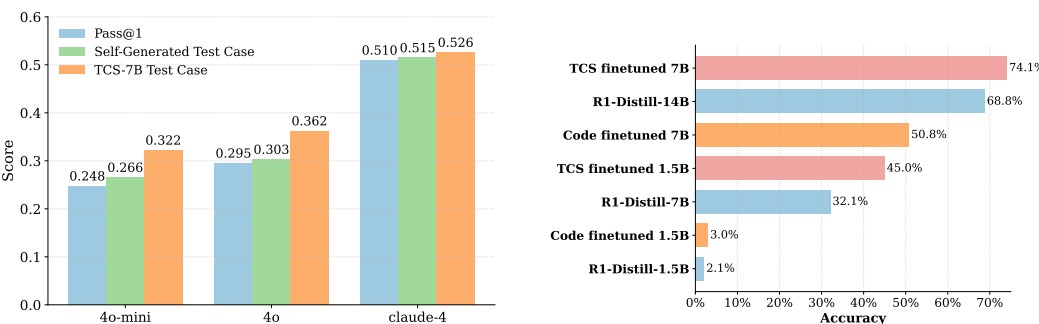

Figure 3: (a) Pass@1 (blue) and BoN (N=8) (green/orange) performance selected by self-generated or TCS-7B-generated test cases for proprietary models in LiveCodeBench. (b) Test case output prediction accuracy.

Our TCS approach demonstrates substantial superiority in code generation, and the use of self-generated test cases further amplifies this advantage. In this section, we conduct a deeper evaluation of the quality of the test cases generated by our model along two dimensions. The first dimension is adversarial effectiveness: whether the generated test cases can reliably expose errors in code. The second dimension is correctness: whether the model, when provided with a test case input, can accurately predict the corresponding output. For the latter, we leverage the test case output prediction task provided by LiveCodeBench.

As shown in Figure 3(a), test cases generated by our TCS-7B model are more effective than those produced by leading proprietary models, and enable these models to achieve superior inference-time scaling when evaluated with our test cases. Figure 3(b) further shows that our TCS-7B model attains the highest accuracy on this benchmark, outperforming the R1-Distill-14B model. Similarly, our TCS-1.5B model achieves substantial gains, surpassing the R1-Distill-7B model. Collectively, these results

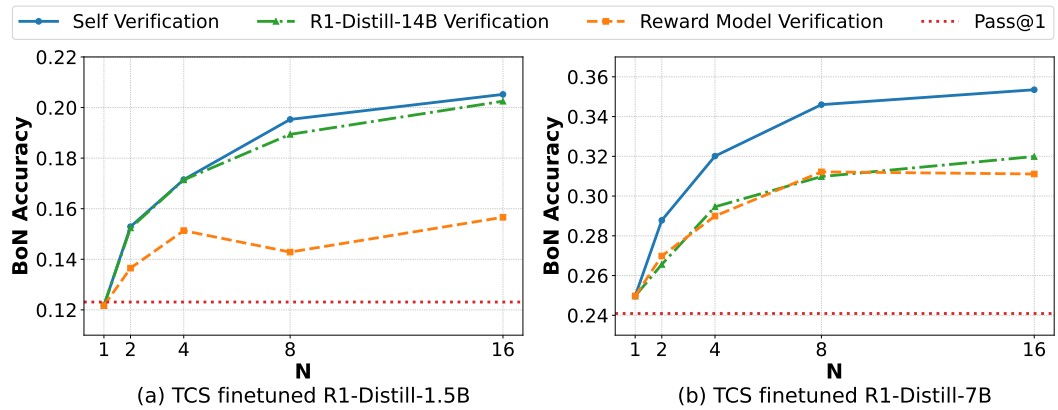

Figure 4: Comparison of different inference-time scaling methods on TACO.

Table 2: Inference-time scaling comparison for the 7B model on TACO with varying reward settings.

| Reward Setting | Method | N=1 | N=2 | N=4 | N=8 | N=16 |
|---|---|---|---|---|---|---|
| only Stage-1-Reward | + public case | 24.23 | 29.89 | 33.00 | 34.89 | 35.83 |
| | + public case & self-gen | 24.23 | 29.68 | 33.40 | 35.80 | 36.70 |
| Two-Stage-Reward | + public case | 24.97 | 28.89 | 32.37 | 33.99 | 36.26 |
| | + public case & self-gen | 24.97 | 29.31 | 33.82 | 37.38 | 39.40 |

demonstrate that our method significantly enhances the reasoning capabilities of code generation models, even on tasks not encountered during training, underscoring its strong generalization ability.

### 4.4 INFERENCE-TIME SCALING COMPARISON

Inference-time computation is a widely adopted strategy to improve answer accuracy. A common approach is to sample multiple candidate answers and use a reward model to rank them, selecting the one with the highest score. For code generation, an alternative test-time scaling approach is to have the LLM generate test cases for self-verification. We use DeepSeek-R1-Distill-Qwen-14B as a strong baseline, applying the same selection logic described in Section 3.3.

Figure 4 demonstrates that our method achieves the strongest inference-time scaling performance. Notably, the TCS-fine-tuned 1.5B model surpasses the much larger 14B model, and the TCS-fine-tuned 7B model shows significant improvements. The reward model's performance fluctuates with increasing sampled candidates, suggesting that excessive candidates may introduce noise and hinder stable scaling. In contrast, self-verification using generated test cases consistently yields robust improvements as $N$ increases, highlighting the unique effectiveness of generated test case selection for scaling code generation models.

### 4.5 EFFECTIVENESS OF TWO-STAGE REINFORCEMENT LEARNING

We hypothesize $R_1^t$ is a lenient training objective, while $R_2^t$ is stricter. To test this, we conduct two experiments: one to see if $R_1^t$ alone suffices, and another to assess the inefficiency of using $R_2^t$ directly.

In the first experiment, we train with only the stage-1 reward under the same computational budget. Table 2 shows the model's limited inference-time scaling and lack of improvement with public test case filtering. Conversely, models with stage-2 reward show clear gains as candidate solutions $N$ increase.

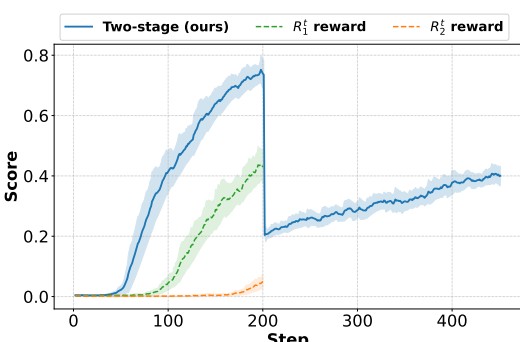

Figure 5: Scores under different training rewards.

In the second experiment, we use a smaller batch size with the 1.5B model to evaluate the direct application of $R_2^t$ before large-scale training. Figure 5 illustrates the necessity of our two-stage design. Applying $R_2^t$ initially requires over 100 training steps to yield meaningful rewards, leading to wasted test case generation samples, as the GRPO algorithm cannot fully utilize all incorrect solutions. In our main experiments, we use a larger batch size and switch to stage 2 at step 200. This transition allows stage 2 with $R_2^t$ to leverage a strong initialization, facilitating continuous improvement. The temporary score drop during the transition indicates a stricter evaluation metric, not a reduction in the model's capability. Once stage 2 starts, $R_2^t$ effectively uses its strong initialization for steady progress.

## 5 Related Work

### 5.1 Code and Test Case Generation Training

Current research on large language models (LLMs) for coding has primarily focused on code generation, with post-training techniques significantly enhancing model capabilities in this area (Hui et al., 2024b; Guo et al., 2024; Jiang et al., 2024b; Fan et al., 2025). However, test case generation—an equally critical aspect of coding proficiency—has received comparatively less attention, especially regarding reinforcement fine-tuning approaches (Yang et al., 2024). Lin et al. (2025) proposed leveraging supervised fine-tuning (SFT) and direct preference optimization (DPO) to improve test case generation abilities. Li et al. (2025) enhanced models' test case output prediction through SFT. Takerngsaksiri et al. (2025) introduced a reinforcement learning framework to optimize the transformation of descriptions into test cases. Recently, CURE Wang et al. (2025) proposed co-evolving coding and unit test generation using RL, but they do not consider the targeted code solution in the test case generation process and overlook the adversarial quality. In contrast to these works, our study is the first to introduce the concept of generating adversarial test cases within the reinforcement learning paradigm for LLMs.

### 5.2 Inference-time Scaling

Directly prompting LLMs to generate responses does not fully exploit their capabilities, particularly for complex code-related tasks. Recent inference-time scaling approaches aim to obtain higher-quality responses during inference (Snell et al., 2024; Wu et al., 2024). A typical strategy is to score multiple candidate responses with a reward model and select the highest-scoring one (Cai et al., 2024; Liu et al., 2025). Beyond these general methods, some techniques introduce inference-time scaling specifically for coding tasks. For instance, Shinn et al. (2023) use feedback from a Python interpreter to iteratively revise and correct generated code until a valid solution is found. CodeT (Chen et al., 2023) and TestChain (Li & Yuan, 2024) build a pipeline to prompt the LLM to generate test cases for self-verification. Other works (Light et al., 2024; Yu et al., 2024; Tang et al., 2024) employ search methods during revision, using test case pass rates as reward signals to guide the LLM toward correct solutions. These approaches typically define a search workflow via prompting and rely on external feedback for candidate selection, without fundamentally improving the model's intrinsic coding ability. In contrast, our method does not depend on a reward model or external signals; instead, it generates adversarial test case samples that can reveal specific code errors, representing a novel and promising direction for inference-time scaling.

## 6 Conclusion

Test case generation is an essential skill for human programmers, occupying a significant portion of coding tasks. However, it has been largely overlooked in current research involving RL for code LLMs. We posit that training models to generate diverse and adversarial test cases can comprehensively enhance overall coding capabilities. To address the existing deficiencies in base models' test case generation abilities, we propose a two-stage RL training framework specifically designed to improve test case generation performance, employing distinct reward functions in each stage. Experimental results demonstrate that our approach significantly enhances both adversarial test case generation and code generation capabilities, ultimately achieving both training-time and inference-time scaling laws. Furthermore, it can benefit leading proprietary models by improving their inference-time scaling

ability and exhibits strong generalization performance on test output prediction tasks. Given that test case generation represents a critical capability within coding, leveraging RL for adversarial test case generation and self-optimization warrants broader exploration.

## ETHICS STATEMENT

This study complies with the ICLR Code of Ethics; all authors have read and agreed to abide by it during the submission process. This work does not involve research with human subjects or the collection or processing of personally identifiable information. All experimental data are drawn from publicly available programming datasets and evaluation platforms (TACO and LiveCodeBench) and are used within the scope of their respective licenses and terms of use.

## REPRODUCIBILITY STATEMENT

We have made every effort to ensure reproducibility and provide actionable guidance and resource references in the main text and appendix. **Algorithms and implementation:** the overall method pipeline is shown in Figure 2; the optimization objective and GRPO training details are given in Eq. equation 2; the code reward and test-case rewards are defined in Eqs. equation 5, equation 6, and equation 7; complete training pseudocode is provided in Algorithm 1. **Data and preprocessing:** implementation details are in Appendix A, which specifies the TACO filtering criteria and the resulting subset of 6,318 training problems, and provides an anonymously downloadable dataset link iclr12230/TACO_Train. The LiveCodeBench evaluation window (August 2024 to February 2025) is also stated in the appendix. **Models and resources:** we release anonymous model weights at iclr12230/TCS_1.5B and iclr12230/TCS_7B. The appendix lists key hyperparameters, including batch size, temperature, maximum response length, GRPO group size, the omission of KL loss, and the use of entropy regularization to maintain exploration. **Inference and evaluation:** we provide full prompt templates for code generation, test case generation, and test case output prediction in the appendix. The Best-of-N settings for TACO/LiveCodeBench, the presence or absence of public cases, and the comparison protocol with reward-model selection can be reproduced as described in the main text. **Environment and compute:** we report staged training and evaluation GPU hours on NVIDIA H100 80GB GPUs in Appendix A.3 and provide the training-step configurations for the baselines and TCS to facilitate budget alignment.

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

# A  IMPLEMENTATION DETAILS

## A.1  DATA

The TACO dataset meets our requirements for a training dataset by providing problems, comprehensive test cases, and submitted solutions (which may not always be correct). The original dataset contains 25,433 problems. To ensure accurate evaluation, we first filter out problems with fewer than 50 test cases, leaving 10,605 problems. We then remove problems without any submitted solutions, as these are necessary for evaluating generated test cases, resulting in 7,918 problems. Next, we verify the submitted solutions against the test cases to ensure the problems are verifiable, since some may be of unverifiable types (e.g., multi-turn input). We retain only those problems for which at least one solution passes all test cases, yielding a final training set of 6,318 problems. The processed data is available at iclr12230/TACO_Train. Following standard practice, we include LiveCodeBench problems from August 2024 to February 2025 in our evaluation (He et al., 2025). Since TACO was released in 2023 while the LiveCodeBench window starts in August 2024, there is a strict chronological separation between our training and evaluation corpora, which rules out contamination from the evaluation problems when constructing the training set. For TACO evaluation, which lacks public test cases, the first evaluation test case is designated as public. When public test cases are available, candidate solutions are filtered to include only those passing these tests before applying inference-time selection methods.

## A.2  EXPERIMENTAL CONFIGURATION

We use verl (Sheng et al., 2024) as our RL framework and adopt the default experimental configuration except for the following modifications:

- batch size: 128
- PPO mini-batch size: 64
- GRPO group size: 16
- temperature (for both training and evaluation): 0.8
- maximum response length: 8192

We do not use the KL loss, and we use the entropy loss to sustain the model's entropy.

For DeepSeek-R1-Distill-Qwen-1.5B, the number of training steps is 200 for stage 1 and 250 for stage 2; the code generation baseline is 450 steps. For DeepSeek-R1-Distill-Qwen-7B, the number of training steps is 40 for stage 1 and 160 for stage 2; the code generation baseline is 200 steps. In both cases, the transition from stage 1 to stage 2 is triggered when the test case generation accuracy on the training batches reaches approximately 0.75.

All RL-based methods (code-only baseline and TCS) are trained under matched step budgets for each model size (450 steps for 1.5B and 200 steps for 7B), ensuring a fair comparison in terms of total RL rollouts.

For inference-time scaling, we set $M = 1$ in all experiments, meaning that each sampled code candidate is paired with exactly one generated test case per inference.

We provide the TCS-finetuned 1.5B model at iclr12230/TCS_1.5B and the 7B model at iclr12230/TCS_7B.

We use TRL (von Werra et al., 2020) as our SFT framework. The SFT baseline uses the same training dataset as TCS. We use DeepSeek-R1-Distill-Qwen-32B to generate 32 samples for each problem. For the code generation task, we retain the samples that pass all the test cases. For the test case generation task, we retain the samples whose outputs align with the ground-truth code. We use default configuration of TRL and set the number of training epochs to 3.

For closed-source model experiments in Figure 3(a), we use the API to generate 8 samples for each problem and use the default API configuration.

To further standardize evaluation and ensure full reproducibility, our supplementary code release includes a unified evaluation harness (scripts/test_case_generation_trigger.sh) that can reproduce all reported test-case-based selection scores by simply setting the environment variables

CODE_MODEL_PATH and TEST_CASE_MODEL_PATH; this script fixes decoding and sampling hyperparameters to those described in this section.

## A.3 COMPUTING RESOURCES

The computing resources used at each stage, based on the NVIDIA H100 80GB, are approximately as follows in terms of GPU hours consumed:

| Model | Training | Evaluation on LiveCodeBench | Evaluation on TACO |
|---|---|---|---|
| R1-Distill-Qwen-1.5B | 720 | 10 | 20 |
| R1-Distill-Qwen-7B | 960 | 16 | 32 |

Note that the times reported here refer to the training or evaluation time for a single run of each model, not the total time across all experiments. The rollout response number for each question is 32 for LiveCodeBench and 16 for TACO. The evaluation time includes both code generation and test case generation. Since the number of code and test case generations is the same, they incur similar computational overhead during evaluation.

## A.4 ADVERSARIAL REWARD COMPUTATION

Each generated test case is evaluated only against its paired program $C$. Take stage 2 for details, we do not aggregate performance across multiple incorrect codes $C_{wrong}$, since the objective is to construct a targeted counterexample for a specific error pattern. As long as a test case successfully exposes the flaw in the paired $C_{wrong}$, it is treated as a valid adversarial example (reward 1), and it is not required to act as a universal counterexample that simultaneously fails other incorrect implementations. Regarding robustness, we distinguish between buffer admission and reward computation: during buffer construction, we filter out syntax errors to ensure basic executability, while at reward time we explicitly treat robustness failures as successes. Concretely, if a generated test case is valid for the oracle solution $C^*$ (i.e., $C^*$ executes without error and produces the expected output) but causes the targeted $C_{wrong}$ to raise a runtime error or timeout, we still assign reward 1, encouraging the model to propose corner cases that reveal both logical and robustness defects (e.g., infinite loops or unhandled exceptions).

# B LIMITATIONS

The primary limitation of our method is that, during inference-time test case generation, it currently produces only a single test case per inference. Generating multiple test cases in a single inference would be more efficient, but this is hindered by the challenge of defining an appropriate reward function: intuitive metrics such as counting correct cases or measuring accuracy rates are susceptible to reward hacking. Investigating reinforcement learning strategies that enable simultaneous generation of multiple test cases thus remains a promising direction. Additionally, our current approach uses a hard switch between training stages; exploring soft switching—i.e., dynamically adjusting the proportion of the two types of test case reward functions—may yield improved results. We did not pursue this primarily due to the high resource demands of RL for LLMs, and therefore prioritized a direct and reliable method. Regarding societal impact, while enhanced LLM coding capabilities can improve productivity, they may also increase risks of misuse, such as in interviews and competitions.

Another potential concern is that, unlike standard RL for code generation which typically assumes only an oracle test suite, our framework additionally requires access to an oracle solution $C^*$. In practice, such solutions are not scarce: large collections of verified code solutions are already routinely used for SFT, and the same resources can support our training setup. For future extensions to settings without executable ground truth, one can replace the execution oracle with LLM-as-a-verifier signals or consistency-based checks across multiple reasoning paths (Zhang et al., 2025; Wang et al., 2023), but we view establishing the methodology under precise execution rewards—as done in this work—as a necessary first step to demonstrate the effectiveness of adversarial test case generation.

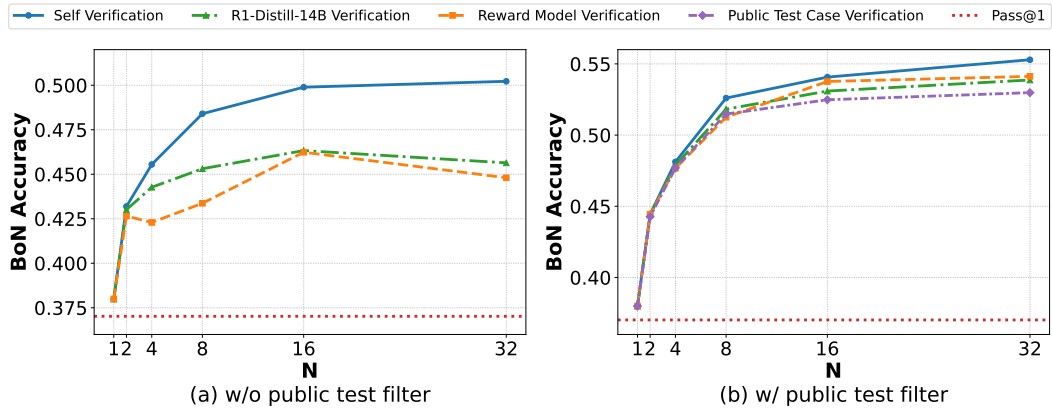

Figure 6: Comparison of different inference-time scaling methods for the TCS-finetuned 7B model on LiveCodeBench, with and without public cases.

## C  ADDITIONAL EXPERIMENTAL RESULTS

### C.1  INFERENCE-TIME SCALING COMPARISON

In this section, we present additional experiments to demonstrate the advantages of our TCS-finetuned 7B model's inference-time scaling ability, even when strong public cases are provided. Live-CodeBench provides around 2.64 public cases for each question and is a relatively strong method for filtering out incorrect code.

Figure 6 (a) shows the inference-time scaling results without public test cases. Our model has a significant advantage over the other representative methods. The other two methods both experience a substantial drop in performance as the number of responses increases. For reward model verification, this means that more responses introduce additional distractors and out-of-distribution (OOD) data, which reduces the probability of making the correct selection. For the more powerful code generation model, Deepseek-R1-Distill-Qwen-14B, since it is not specifically trained for adversarial test case generation, it is more likely to produce meaningless or even incorrect test cases, causing some incorrect responses to pass more generated test cases. Therefore, these two baseline methods do not exhibit the inference-time scaling law when there are no manually constructed strong public test cases. In contrast, our TCS-finetuned model demonstrates strong robustness and consistently exhibits the inference-time scaling law.

Figure 6 (b) shows that when strong public test cases are provided, both methods consistently improve as the number of responses increases. However, our method has a clear advantage over the other two comparison methods. In future automated coding scenarios, we do not want code agents to rely on high-quality test cases written by humans, so achieving a consistent scaling law without human-provided test cases is especially important. Of course, when some test cases are provided, our method can also generate even more complex test cases.

### C.2  RL TRAINING WITH ONLY THE CODE GENERATION TASK

In this section, we explore whether RL training exclusively on our collected data for code generation, rather than for test case generation, can indirectly improve the model's capability in generating test cases. The experimental setup in Table 3 shows that the accuracy gains in test case generation from RL with code training are limited. This result highlights the necessity of treating test case generation as an independent task that requires dedicated RL to achieve substantial improvements.

### C.3  DETAILED RESULTS IN TACO EVALUATION

There are five difficulty levels in the TACO evaluation. In this section, we present the detailed results for each difficulty level. (TC) means using test cases generated by the model itself to choose the best solution. Our results demonstrate that finetuning with our approach yields significant

Table 3: Performance on TACO and LiveCodeBench

| Model | TACO | | LiveCodeBench | |
|---|---|---|---|---|
| | w/o public | w/ public | w/o public | w/ public |
| **DeepSeek-R1-Distill-Qwen-1.5B** | | | | |
| Base Model | 5.63 | | 14.38 | |
| + Reward Model | 12.70 | 14.60 | 23.30 | 30.11 |
| + Self-Generated Test Cases | 5.78 | 13.91 | 22.00 | 30.09 |
| RL using Code Training | 11.16 | | 18.53 | |
| + Reward Model | 15.87 | 22.54 | 20.79 | 34.50 |
| + Self-Generated Test Cases | 11.92 | 21.68 | 21.05 | 33.90 |
| **RL using TCS Training** | 12.31 | | 20.63 | |
| + Reward Model | 15.66 | 23.17 | 24.01 | 37.99 |
| **+ Self-Generated Test Cases** | **20.52** | **25.18** | **27.47** | **38.90** |

improvements in code generation performance, surpassing both the base model and the base model with self-verification. The benefits of our method become increasingly evident as task difficulty rises. Additionally, leveraging self-generated test cases delivers further inference-time scaling gains, with particularly notable improvements on more challenging tasks. These findings highlight the adaptability of our approach to difficult problems and its potential applicability to stronger models and more complex tasks.

| | **EASY** | **MED** | **MED_HARD** | **HARD** | **V_HARD** | **TOTAL** |
|---|---|---|---|---|---|---|
| **R1-7B** | 34.87 | 24.78 | 13.09 | 2.75 | 0.72 | 14.36 |
| **R1-7B(TC)** | 43.71 | 32.32 | 17.2 | 4 | 1.53 | 18.67 |
| **TCS-7B** | 46.11 | 37.53 | 24.44 | 10.56 | 6.66 | 24.09 |
| **TCS-7B(TC)** | 58.88 | 50.55 | 34.43 | 25.31 | 12.76 | 35.35 |

Table 4: Performance comparison across different difficulty levels and total scores for various models.

## C.4 JOINT VS. DECOUPLED TRAINING OF CODE AND TEST CASE POLICIES

To further disentangle the contributions of our TCS framework from generic RL effects, we additionally compare joint training against two decoupled baselines: (i) RL for code generation only ("Code RL") and (ii) RL for test case generation only ("Test RL", referred to as Model 3). Table 5 summarizes the training-time and inference-time performance (without public test cases) of these variants on TACO and LiveCodeBench using the R1-Distill-Qwen-1.5B backbone.

Table 5: Comparison of joint versus decoupled training for code and test case policies (R1-Distill-Qwen-1.5B, without public test cases). "TC" denotes selection with self-generated test cases.

| Model | TACO (w/o pub) | LiveCodeBench (w/o pub) |
|---|---|---|
| R1-Distill-Qwen-1.5B | 5.63 | 14.38 |
| + TC | 5.78 | 22.00 |
| Code RL only | 11.16 | 18.53 |
| + TC | 11.92 | 21.05 |
| Test RL only (Model 3) | 7.85 | 15.95 |
| + TC | 15.35 | 23.26 |
| **RL using TCS Training** | **12.31** | **20.63** |
| **+ TC (self-generated)** | **20.52** | **27.47** |
| + TC from Model 3 | 18.79 | 25.94 |

These results clarify the respective roles of code and test case RL. First, Code RL only substantially improves pass@1 over the base model but yields weak gains from self-generated test case selection,

indicating that it does not learn to produce sufficiently discriminative test cases for inference-time filtering. In contrast, Test RL only (Model 3) learns a stronger verifier: although its direct pass@1 is modest, its self-generated test cases noticeably enhance inference-time selection. However, the joint TCS training achieves the best of both worlds: it delivers the strongest training-time performance and the largest inference-time improvements under self-generated test case selection. Moreover, using Model 3's test cases to filter TCS-generated code still underperforms TCS's own self-verification, suggesting a synergistic self-play effect—as the solver improves, it produces harder negative examples that in turn drive a more powerful test case generator.

### C.5 CODERM-8B BASELINE COMPARISON

We additionally compare our TCS-finetuned 7B model with CodeRM-8B, a unit-test-generation model obtained by supervised fine-tuning (SFT) Llama-3.1-8B on test cases synthesized by a larger Llama-3.1-70B teacher. To ensure a fair comparison, we use the same inference-time budget $N$ for all methods when sampling code candidates and associated test cases. The pass@1 results on TACO and LiveCodeBench are summarized in table 6.

Table 6: Comparison of different inference-time selection methods for the TCS-finetuned 7B model, using either a reward model, CodeRM-8B-generated tests, or self-generated tests.

| Method | TACO | | LiveCodeBench | |
|---|---|---|---|---|
| | w/o pub | w/ pub | w/o pub | w/ pub |
| RL using TCS-7B Training | 24.09 | 37.03 | 37.03 | 37.03 |
| + Reward Model | 31.11 | 37.67 | 43.01 | 53.40 |
| + CodeRM-8B Generated Test Cases | 23.67 | 35.90 | 41.70 | 51.49 |
| + Self-Generated Test Cases | **35.35** | **39.40** | **48.79** | **54.75** |

We observe that CodeRM-8B's test cases yield weaker filtering performance than both our reward-model baseline and our self-generated tests. We attribute this to three factors. First, the base model underlying CodeRM-8B is substantially smaller and less capable than the TCS-7B backbone, limiting the quality of its generated tests. Second, CodeRM-8B relies purely on SFT, whereas our method uses RL; as shown in Table 1, under the same base model RL consistently outperforms SFT for code-related tasks. Third, CodeRM-8B acts as an external verifier, while TCS jointly trains the solver and verifier using a policy-aligned buffer, so the test generator is explicitly optimized to expose the current failure modes of the solver. This tight alignment between policies leads to more targeted and effective bug discovery than a generic external verifier.

## D ADVERSARIAL GENERATED TEST CASE EXAMPLE

A correct response in training stage two is to generate a test case that the ground truth solution can pass, but the generated incorrect solution will fail. Figure 7 provides an example. The generated test case has a valid input, and the correctness of its output is verified by the ground truth code. This test case can efficiently reveal the errors in the generated code, so it is the expected adversarial test case. The following is a snippet from the complete reasoning process that shows the motivation behind constructing this test case:

> "...But maybe that's incorrect. For example, if the last known draw was at x and the next known is y where y > x, but in between, maybe there were more draws possible? Or perhaps it's only considering overlapping ranges.
> Wait, maybe in some cases, the code is not correctly handling multiple overlapping intervals. For example, between two known points, the number of possible draws could be more than what the code calculates.
> Let me think of a test case that would expose an error in the code..."

To better demonstrate that our TCS-finetuned model can generate high-quality, challenging, and adversarial test cases, we present a generated test case that is able to filter out incorrect code that already passes all the public test cases, as shown in Figure 8. The problem is a difficult challenge

**Problem** You still have partial information about the score during the historic football match. You are given a set of pairs $(a_i, b_i)$ in chronological order (time increases), indicating that at some point during the match the score was "$a_i : b_i$". The last pair corresponds to the end of the match. What is the largest number of times a draw could appear on the scoreboard?

**Generated Code**

```
n = int(input())
known = [tuple(map(int, input().split()))
for _ in range(n)]
total = 1  # initial 0:0
for i in range(n-1):
    prev_a, prev_b = known[i]
    curr_a, curr_b = known[i+1]
    s_prev = max(prev_a, prev_b)
    e_curr = min(curr_a, curr_b)
    if prev_a == 0 and prev_b == 0:
        count = e_curr
    else:
        if e_curr >= s_prev:
            count = e_curr - s_prev + 1
        else:
            count = 0
    total += count
print(total)
```

**Ground Truth Code**

```
n = int(input())
(a0, b0) = (0, 0)
ans = 1

for i in range(n):
  (a, b) = map(int, input().split())
  ans += max(min(a, b) - max(a0, b0), 0)
  if min(a, b) - max(a0, b0) >= 0 and
a0 != b0:
    ans += 1
  (a0, b0) = (a, b)

print(ans)
```

**Generated Test Case**

```
Input: 3\n0 0\n2 2\n3 3
Generated Code Output: 5
Ground Truth Output: 4
```

Figure 7: A generated test case that receives a reward in training stage two.

**Generated Code**

```python
def main():
    n, m, l = map(int, sys.stdin.readline().split())
    A = list(map(int, sys.stdin.readline().split()))
    B = list(map(int, sys.stdin.readline().split()))
    C = list(map(int, sys.stdin.readline().split()))
    A_sorted = tuple(sorted(A))
    B_sorted = tuple(sorted(B))
    C_sorted = tuple(sorted(C))
    @lru_cache(maxsize=None)
    def can_win(tak_hand, aoki_hand, table, is_takahashi_turn):
        if is_takahashi_turn:
            for x in tak_hand:
                new_table = list(table) + [x]
                new_table_sorted = sorted(new_table)
                for y in new_table_sorted:
                    if y < x:
                        new_t = []
                        for z in new_table_sorted:
                            if z != y:
                                new_t.append(z)
                        new_t = tuple(new_t)
                        new_t_hand = tuple([z for z in tak_hand if z != x])
                        if not can_win(new_t_hand, aoki_hand, new_t, False):
                            return True
            return False
        else:
            for x in aoki_hand:
                new_table = list(table) + [x]
                new_table_sorted = sorted(new_table)
                for y in new_table_sorted:
                    if y < x:
                        new_t = []
                        for z in new_table_sorted:
                            if z != y:
                                new_t.append(z)
                        new_t = tuple(new_t)
                        new_a_hand = tuple([z for z in aoki_hand if z != x])
                        if not can_win(tak_hand, new_a_hand, new_t, True):
                            return True
            return False
    result = can_win(A_sorted, B_sorted, C_sorted, True)
    if result:
        print("Takahashi")
    else:
        print("Aoki")
if __name__ == "__main__":
    main()
```

**Problem**

Takahashi and Aoki will play a game using cards with numbers written on them. Initially, Takahashi has N cards with numbers $A_1, \ldots, A_N$ in his hand, Aoki has M cards with numbers $B_1, \ldots, B_M$ in his hand, and there are L cards with numbers $C_1, \ldots, C_L$ on the table. Throughout the game, both Takahashi and Aoki know all the numbers on all the cards, including the opponent's hand. Starting with Takahashi, they take turns performing the following action:
• Choose one card from his hand and put it on the table.
Then, if there is a card on the table with a number less than the number on the card he just played, he may take one such card from the table into his hand. The player who cannot make a move first loses, and the other player wins. Determine who wins if both players play optimally. It can be proved that the game always ends in a finite number of moves.

**Public Test Case**      Pass All

```
Input:  1 1 2\n2\n4\n1 3
Output: Aoki

Input:  4 4 4\n98 98765 987654
987654321\n987 9876 9876543
98765432\n123 12345 1234567
123456789
Output: Takahashi

Input:  1 1 8\n10\n10\n1 2 3 4 5 6 7
8
Output: Aoki
```

**Generated Test Case**      Fail

```
Input: 2 1 2\n2 3\n1\n4 5
Expected Output: Takahashi
Generated Code Output: Aoki
```

Figure 8: A generated test case that can identify an error in the generated code, even though it passes all the public test cases.

from the AtCoder platform[1] with a difficulty score of 500 and relatively strong public test cases. During inference, we generate 32 responses, and three of them pass all the public test cases. With our generated test case, we can successfully filter out two solutions with hard-to-detect errors (one is shown in the figure), ultimately selecting the only correct answer. We provide the full reasoning process for this generated test case in Listing 1.

# E   PROMPT TEMPLATE

We provide all the templates used for code generation, test case generation, and test case output prediction in this section.

**Code Generation Prompt**   This prompt is the same as the one used in the LiveCodeBench benchmark for code generation.

---
[1]https://atcoder.jp/contests/abc380/tasks/abc380_f

**Prompt Template for Code Generation**

```
You will be given a question (problem specification) and will
generate a correct Python program that matches the specification
and passes all tests.
{problem}
Read the inputs from stdin solve the problem and write the answer
to stdout (do not directly test on the sample inputs). Enclose your
code within delimiters as follows. Ensure that when the python
program runs, it reads the inputs, runs the algorithm and writes
output to STDOUT.
```python
# YOUR CODE HERE
```

**Test Case Generation Prompt** There are four configurable items in the test case generation prompt template. For each **problem**, we require the model to generate a test case that can reveal potential errors in a specific **code**. We also provide the format of example **test cases** for reference, but the generated test case should be different from these examples. To improve the diversity of the prompts, we define four **types** of test cases and randomly sample one when constructing a test case:

- **basic**: basic test case that validates core functionality with simple, straightforward inputs.
- **edge**: edge case that tests boundary values and constraint limits (minimum/maximum allowed values)
- **corner**: corner case with unusual inputs like empty collections, single elements, or patterns that might break naive solutions
- **performance**: performance test with large inputs approaching the problem's limits to evaluate solution efficiency

**Prompt Template for Test Case Generation**

```
You are an expert TEST CASE GENERATOR for programming competitions.
Your ONLY task is to generate ONE TEST CASE of a specific type, NOT
to solve the problem or write any implementation code.

Follow these strict guidelines:
1. DO NOT write any solution code in any programming language.
2. DO NOT attempt to fix or improve the solution.
3. FOCUS EXCLUSIVELY on generating a {test_case_type} that can
reveal flaws or confirm correctness.
4. The test case you generate MUST NOT be identical to any Example
Test Case provided in the problem statement.

You have been provided with:
* Problem Description:
{problem}
* A SOLUTION CODE THAT MAY CONTAIN LOGIC ERRORS:
```python
{code}
```

Your goal is to create a test case that is valid under the problem
constraints and is likely to expose incorrect behavior in the
provided code if such errors exist.

For the test case you create, provide:
* The test input exactly as it would be fed to a program.
* The expected output that a correct solution should produce.
* A brief explanation of what aspect this test case is verifying or
how it could reveal flaws.
```

```
Ensure the output follows the *Expected Output Format* structure
provided. You must enclose the output in a ```json``` block to
facilitate easy extraction and processing.

Expected Output Format:
```json
{expected_output}
```
You must ONLY provide ONE {test_case_type}. Ensure your test case
is valid according to the problem constraints.
```

**Test Case Output Prediction Prompt**    In this prompt, we strictly require the model to output the prediction in the expected format to ensure that the answer can be correctly extracted.

**Prompt Template for Test Case Output Prediction**

```
You are an expert TEST CASE GENERATOR for programming competitions.
Now you are given a test case input, and you need to infer what its
output should be.
* Problem Description:
{problem}
* Test Case Input:
{input}
Please strictly follow the format below to return your answer:
Expected Output Format:
```json
{expected_output}
```
Remember you task is predict the output of the test case and you
need to return the output in the format of json.
```

1188
1189

### Listing 1: An Example of the Reasoning Process for Test Case Generation

1190
1191

Assistant: Okay, I need to create a test case for this problem. Let me first understand the
    problem thoroughly.

1192
1193
1194
1195
1196

The problem is about a game between Takahashi and Aoki using cards. They take turns playing
    cards from their hands onto the table, and during their turn, they can also take a card
    from the table if it's smaller than the card they just played. The player who can't make
    a move loses.

The task is to determine who wins if both play optimally.

1197
1198

The constraints are that N + M + L ≤ 12, which is a small number, so the solution can afford
    to explore all possible states without performance issues.

1199
1200
1201

The provided code uses memoization with functools.lru_cache to cache the results of the game
    states. It represents the state as the remaining cards in Takahashi's hand, Aoki's hand,
    the sorted table cards, and whose turn it is.

1202
1203

Now, I need to generate a test case that's valid, meaning the inputs must satisfy the
    constraints: N, M, L are at least 1, and their sum is ≤12. The test case should not be
    identical to any example provided.

1204
1205

Looking at the sample inputs, they all have N, M, L as 1, 1, 2 or similar. To create a new
    test case, I can choose different values.

1206
1207

Let me think of a case where the outcome is Takahashi wins. Maybe a simple scenario where
    Takahashi can always force a win.

1208
1209

For example, let's take N=2, M=1, L=3. A = [1, 2], B = [3], C = [4,5,6].

1210
1211

In this case, Takahashi plays 1, then Aoki plays 3, but the table has 1, so Aoki can take 1.
    Then Takahashi plays 2, and the table has 2, so he can take 1 again. Aoki has no moves
    left, so Takahashi wins.

1212
1213

Wait, but I need to ensure that the test case is valid and that the code can handle it
    correctly.

1214
1215

Alternatively, perhaps a case where the table initially has a card that can be taken by both
    players.

1216
1217

Wait, but the initial state includes the table cards. So, for example, if the table has a card
     that is smaller than both players' cards, it can affect the outcome.

1218
1219

Let me create a test case where the initial table has a card that allows Takahashi to take it
    on his first move, putting Aoki in a bad position.

1220
1221

For example:

N=1, M=1, L=1
A = [2]
B = [3]
C = [1]

1222
1223
1224

So, Takahashi plays 2, then checks the table for cards <2. The table has 1, so he takes it.
    Now, his hand is empty, Aoki's hand is [3], and the table has 3.

1225
1226

Aoki's turn: he plays 3, checks the table for <3, which is none, so he can't take anything.
    Now, Takahashi has no moves because he has no cards left, so Aoki wins.

1227
1228

Wait, but in this case, Takahashi took the 1, so after his move, the table has 3. Aoki plays
    3, no cards to take, so Aoki can't make a move? Wait, no, Aoki plays 3, and then checks
    if there's a card on the table less than 3. The table has 3, so no. So Aoki's move is
    valid, and after that, the table is empty. Then Takahashi has no cards left, so he can't
    play anything and loses.

1229
1230
1231
1232

So the output should be Aoki.

1233

But I'm not sure if this test case is correctly capturing any specific scenario. Maybe I
    should create a test case where Takahashi can win.

1234
1235

Alternatively, let's consider a case where Takahashi has a higher card than all on the table,
    and Aoki can't respond.

1236
1237

For example:

N=1, M=1, L=1
A = [3]
B = [2]
C = [1]

1238
1239
1240
1241

Takahashi plays 3, takes 1 if possible. Then the table has 2. Aoki plays 2, takes 1 if
    possible. Then Takahashi has no cards left, so Aoki wins again.

Hmm, maybe it's tricky to find a test case where Takahashi wins. Perhaps the provided code
    handles all cases correctly, but I need to ensure that the test case is valid.

Wait, the sample input 1 has N=1, M=1, L=2, and the output is Aoki. So perhaps a test case
    where the outcome is Takahashi can be created.

Let me try:

N=1, M=1, L=1
A = [1]
B = [2]
C = [3]

Takahashi plays 1, checks the table for <1, which is none, so he can't take anything. Now, the
    table has 3. Aoki's turn: he plays 2, checks the table for <2, which is none, so he can'
    t take anything. Now, Takahashi has no cards left, so he can't play, and Aoki wins.
    Output is Aoki.

Another test case:

N=2, M=1, L=2
A = [2, 3]
B = [1]
C = [4,5]

Takahashi can play 2, take 4 if possible? Wait, 4 is not less than 2, so he can't take it. So
    the table has 2 and 4 and 5. Aoki plays 1, takes 2 if possible. Since 2 >1, he can't take
    it. So Aoki's move is valid, and the table has 4 and 5. Then Takahashi plays 3, checks
    for <3, which is none, so he can't take anything. Aoki has no cards left, so he can't
    play and loses. Takahashi wins.

Wait, but in this case, after Takahashi plays 2, the table has 2,4,5. Aoki plays 1, and since
    1 <2, he can take 2. So Aoki takes 2, adds it to his hand, and the table now has 4 and 5.

Then Takahashi's turn: he has 3 left. He plays 3, checks the table for <3, which is none, so
    he can't take anything. Aoki has no cards left, so he can't play and loses. So Takahashi
    wins.

So the test case would be:

Input:
2 1 2
2 3
1
4 5

Output: Takahashi

This should be a valid test case, and it's different from the samples provided.

So, the test case would be:

N=2, M=1, L=2
A = [2,3]
B = [1]
C = [4,5]

The expected output is Takahashi.

This test case should be valid, and it's different from the samples, so it's a good candidate
    to test the code.
</think>

```json
[
  {
    "input": "2 1 2\n2 3\n1\n4 5",
    "output": "Takahashi"
  }
]
```