# OpenReview forum: "Adversarial Test Case Generation via Reinforcement Learning Extends Scaling Laws"
_ICLR.cc/2026/Conference — Submitted to ICLR 2026_

### Official Review · Reviewer_j3Nq · 2025-10-27

**Soundness:** 2
**Presentation:** 2
**Contribution:** 1
**Rating:** 2
**Confidence:** 4

**Summary:**

The paper proposes a two-stage reinforcement learning framework for code and test cases generation. Their main contribution is the integration of RL into the test case generation, which is utilized to generate "adversarial" test cases and improve code quality.

**Strengths:**

The application of RL to the problem of adversarial test case generation is novel in the context of LLM coding.

When done properly, an adversarial test generator can provide a continuous mechanism for self-correction and iterative improvement.

The presented framework is well described, and the flow of the methodology is clear.

**Weaknesses:**

**Fundamental Flaws in RL Reward Function Design**

The methodological execution of the RL framework appears fundamentally flawed due to the definition of the primary reward signals.

Reward $R_1$ Issue: The reward $R_1$ is defined as $1$ if the execution of the target code solution $C*$ with the generated input $I_g$ matches the generated output $O_g$. Since $C^*$ is the correct solution, any valid input $I_g$ will produce a valid output $O_g$. This means a randomly generated input $I_g$ would also achieve $R_1=1$ with high probability. This definition fails to reward intelligent search or adversarial quality.

Adversarial Reward Issue: The adversarial reward, which utilizes the incorrect code $C_{wrong}$, suffers from the same problem. A random search over the input space that simply evaluates inputs on $C^*$ and then checks $C_{wrong}$ would likely achieve a positive reward frequently, making the learned RL policy potentially no better than random exploration.

Missing Justification: The paper fails to provide a compelling argument for the advantage of a learned RL policy over a simple random search over the input space combined with a standard pass/fail evaluation. This lack of a random baseline comparison undermines the entire RL methodology.

**Insufficient and Unclear Evaluation**

The experimental design and reporting lack the necessary rigor to support the paper's claims:

Missing Definition of Key Metrics: The paper must explicitly define how key evaluation metrics are calculated:

* The definition of the pass@1 score is missing.

* The calculation method for the Best-of-N (BoN) score (as noted in the second and third rows of tables) is not provided.

Ambiguity in Results Presentation: For Figure 3(a) ("Pass@1 and BoN (N=8) performance"), it is ambiguous whether the single bar presented for each model/baseline is the average over both metrics, which would be an inappropriate aggregation. The authors need to clarify this presentation.

Unclear Accuracy Calculation: The method for calculating the "accuracy" for the results presented in Figure 3(b) is not defined and requires detailed expansion.

Statistically Insignificant Sample Size: The use of only 16 samples for TACO and 32 for LiveCodeBench is severely limited and insufficient to derive robust conclusions about model performance or the generalization of the test case generator.

Lack of Statistical Rigor: None of the results presented include standard deviations ($\pm$ std) or confidence intervals. This omission prevents any assessment of the statistical relevance of the findings and suggests the experiments may have been conducted with a single trial, rendering the claims of performance improvement tenuous.

**Questions:**

Please see weaknesses.

Can you add clarity to the score/accuracy definitions?

Where the evaluation done only on 1 trail?

Why only 16 samples for TACO and 32 for LiveCodeBench where used?

---

> ### Author Response · Authors · 2025-12-03
>
> We thank the reviewer for their time and comments. We notice there are some fundamental misunderstandings regarding our task definition and experimental setup, which we clarify below.
>
> **Q1:** *RL Reward Function Design (Fundamental Misunderstanding).*
>
> A1: We respectfully point out a misunderstanding regarding our generation task. The reviewer states that "any valid input $I_g$ will produce a valid output $O_g$." This assumes the model only generates the input. **Clarification**: Our model generates a **complete test case tuple** $(I_g, O_g)$, consisting of both the input and the expected output.
>
> - **Why Reward $R_1^t$ is non-trivial:** The reward $R_1^t$ requires that the execution result of the ground truth code on the generated input ($Exec(C^*, I_g)$) must match the **model-generated output ($O_g$)**. If the model generates a valid input but fails to calculate the correct output logic, the reward is 0.
> - **Why Random Search is insufficient:** A simple random search over the input space can only provide $I_g$. It cannot provide the corresponding ground-truth output $O_g$.
>
> ---
>
> **Q2:** *Clarification on Sample Size.*
>
> **A2:** There appears to be a significant confusion regarding the dataset size. We strictly followed standard benchmarks: the **TACO test set contains 1,000 problems**, and the **LiveCodeBench test set contains 279 problems**. The numbers "16" and "32" mentioned in the paper refer to the **number of candidate samples ($N$) generated per problem**  during inference, **NOT** the number of problems in the test set. Sampling $N=16$ or $N=32$ candidates per problem is a standard setting in code generation literature to estimate expected performance [1]. Therefore, our evaluation is conducted on a large-scale set of 1,279 problems, ensuring statistical robustness.
>
> ---
>
> **Q3:** *Definitions of Metrics (Pass@1, Best-of-N, and Accuracy).*
>
> **A3:** Pass@1 and Best-of-N are standard metrics in Large Language Model reasoning tasks. The former represents the average accuracy on the dataset when the model generates a single response. Best-of-N represents the average accuracy on the dataset where the model generates $N$ responses, and the optimal one is selected via a specific strategy (e.g., our self-generated test cases) to be submitted as the answer.
>
> Regarding Figure 3(a), the results are not an average of Pass@1 and BoN. Instead, the bars represent distinct metrics for different settings: the blue bar represents the model's baseline Pass@1 score, the green bar denotes the BoN score when using self-generated test cases for selection, and the orange bar indicates the BoN score when using test cases generated by our trained TCS-7B model for selection.
>
> Regarding the Accuracy in Figure 3(b), it simply refers to the correctness rate. Given a code problem and a specific test input, it calculates the probability that the output inferred by our model matches the ground-truth output.
>
> **Revision:** We have added a detailed explanation of the metric definitions in Figure 3(a) to the revised manuscript.
>
> ---
>
> **Q4:** *Statistical Rigor and Single Trial.*
>
> **A4:** Reporting results from a single training run is the standard practice in the LLM field due to the prohibitive computational costs [1,2,3]. All experiments in this paper are estimated to have cost over **USD 100,000**. Retraining multiple times to obtain standard deviations is computationally infeasible. While we train once, the evaluation itself is statistically robust because the reported metrics are aggregated over **1,279 total test problems** with multiple stochastic samples ($N=16/32$) per problem.
>
> ---
>
> [1] He et al. "Skywork Open Reasoner 1 Technical Report." arXiv preprint arXiv:2505.22312 (2025).
>
> [2] Lin et al. "Learning to Solve and Verify: A Self-Play Framework for Code and Test Generation." arXiv preprint arXiv:2502.14948 (2025).
>
> [3] Shao et al. "DeepSeekMath: Pushing the Limits of Mathematical Reasoning in Open Language Models." arXiv preprint arXiv:2402.03300 (2024).

---

### Official Review · Reviewer_B6xj · 2025-10-27

**Soundness:** 3
**Presentation:** 3
**Contribution:** 2
**Rating:** 4
**Confidence:** 4

**Summary:**

This paper targets the under-explored problem of learning to generate adversarial test cases for code evaluation. The authors propose a two-stage RL framework (TCS): Stage I learns to produce correct and non-duplicative tests; Stage II learns to produce adversarial tests that distinguish incorrect solutions. A policy-aligned buffer keeps the test generator focused on the solver’s current error modes. At inference time, the model generates tests for each candidate program and selects the candidate that passes the most tests (self-generated tests, majority vote). Experiments on TACO and LiveCodeBench with 1.5B/7B backbones show improvements over SFT and simple self-verification baselines and illustrate positive scaling with more trajectories.

**Strengths:**

1) Timely focus: Training a model to author adversarial tests addresses a real bottleneck in sample-and-verify pipelines for code generation.
2) Simple and practical design: The two-stage reward and rolling buffer are easy to implement and reason about.
3) Clear deployment story: The inference-time selection via self-generated tests is straightforward and actionable.
4) Empirical signal: Consistent gains over SFT/simple self-verification across two model sizes; reasonable data-scaling trends.

**Weaknesses:**

1) Missing head-to-head comparison with verifier-based methods
   The paper’s inference-time scheme:“generate tests → execute → vote”—aligns closely with the verifier paradigm. Strong baselines in this space are independent verifier models (e.g., CodeRM-8B) that are trained specifically to write high-quality tests and discriminate with small test budgets.
2) Data and comparability details
   Training/evaluation are focused on Python/contest-style tasks. Generalization to other languages and more heterogeneous settings is not validated. If benchmarks are patched or randomized, a standardized evaluation harness (fixed temperatures, retries, timeouts, tool stacks) is needed to ensure cross-paper comparability.
3) Leakage and template-dependence checks
   If public exemplars/writeups or replays are used, stronger evidence of near-duplicate filtering, time splits, and template obfuscation is needed to ensure adversarial tests are not memorized patterns.

**Questions:**

1) Can you provide same-candidate-pool, same N×M budget comparisons against CodeRM-8B , and report the minimum M required to reach a target accuracy?
2) Will you release a standardized evaluation harness with fixed decoding settings to ensure cross-paper comparability?
3) Do you have near-duplicate/time-split/template-obfuscation analyses to rule out leakage or pattern memorization in adversarial tests?

---

> ### Author Response · Authors · 2025-12-03
>
> Thank you for your constructive comments and suggestions. Below please find our response to your questions.
>
> **Q1:** *Add CodeRM-8B [1] baseline.*
>
> **A1:** CodeRM-8B is a unit test generation model fine-tuned (SFT) on Llama-3.1-8B using data synthesized by Llama-3.1-70B. To ensure a fair comparison, we used the same budget ($M=1$) to sample the test cases. The results are as follows:
>
> |                                 | TACO (w/o pub & w/ pub) | LiveCodeBench (w/o pub & w/ p) |
> | ------------------------------- | ----------------------- | ------------------------------ |
> | RL using TCS-7B Training        | 24.09                   | 37.03                          |
> | +Reward Model                   | 31.11 & 37.67           | 43.01 & 53.40                  |
> | +CodeRM-8B Generated Test Cases | 23.67 & 35.90           | 41.70 & 51.49                  |
> | +Self-Generated Test Cases      | 35.35 & 39.40           | 48.79 & 54.75                  |
>
> We observe that CodeRM-8B's performance lags significantly behind our method and is even weaker than our Reward Model baseline. We attribute this to three key factors:
>
> * **Base Model Capability:** Its base model capability is significantly weaker than the base model used in our approach.
>
> * **Training Paradigm (SFT vs. RL):** CodeRM relies solely on SFT. As shown in Table 1 of our paper, under the same base model, our proposed RL method significantly outperforms SFT methods.
>
> * **Policy Alignment:** CodeRM is an external verifier. In contrast, TCS is trained with a **policy-aligned buffer**, ensuring the test case generator is specifically optimized to expose the current error modes ("weaknesses") of the solver. This synergy allows for more efficient, targeted bug detection than a generic external model.
>
> **Revision:** We have added Appendix C.5 to include the CodeRM-8B baseline comparison and the corresponding analysis.
>
> ---
>
> **Q2:** *standardized evaluation harness.*
>
> **A2:** Yes, we ensure full reproducibility. Our supplementary code submission includes a standardized evaluation harness. Specifically, the script `scripts/test_case_generation_trigger.sh` enables the reproduction of all reported scores by simply defining the `CODE_MODEL_PATH` and `TEST_CASE_MODEL_PATH`. This harness uses fixed decoding settings (as detailed in Appendix A.2 ) to ensure fair cross-paper comparability.
>
> **Revision:** We have updated Appendix A.2 to explicitly reference the standardized evaluation script included in the supplementary material.
>
> ---
>
> **Q3:** *Leakage & Memorization.*
>
> **A3:** Strict Time-Split Analysis: We fine-tuned our model using the TACO dataset (published in 2023), whereas our evaluation strictly employs the LiveCodeBench window from August 2024 to February 2025. This strict chronological gap guarantees that the evaluation problems did not exist during the creation of our training data, effectively ruling out data contamination issues.
>
> Generalization Evidence: As demonstrated in Figure 3 and Section 4.3, our TCS-7B model effectively generates test cases to verify code solutions from closed-source models (e.g., GPT-4o, Claude) on these unseen problems. The model's ability to identify errors in OOD (out-of-distribution) solutions generated by external models confirms that it has learned robust adversarial verification logic rather than memorizing specific problem-solution patterns.
>
> **Revision:** We have reinforced the description of the data time-split in Appendix A.1 to clarify the strict chronological separation.
>
> ---
>
> **Q4:** *Generalization to other languages.*
>
> **A4:** The method and algorithm proposed in this paper are **language-agnostic**. The sole reason for focusing on Python for code generation in this work is that the current mainstream training datasets and benchmarks in the code generation field are predominantly centered around Python.
>
> ---
>
> [1] Ma et al. "Dynamic Scaling of Unit Tests for Code Reward Modeling." arXiv preprint arXiv:2501.01054 (2025).

---

### Official Review · Reviewer_ChCt · 2025-11-01

**Soundness:** 2
**Presentation:** 3
**Contribution:** 2
**Rating:** 4
**Confidence:** 3

**Summary:**

The paper proposes Test Cases Scaling (TCS), a two-stage reinforcement learning framework that trains an LLM not only to solve programming problems but also to generate adversarial test cases that expose errors in incorrect code. Stage-1 uses a “valid-but-non-copied” reward $R_{t1}$ to teach the model to produce correct, non-example test cases; Stage-2 switches to an adversarial reward $R_{t2}$ that grants a reward when the ground-truth solution passes but an incorrect solution fails on the generated case. The framework also maintains a policy-aligned buffer of recent code outputs to construct dynamic test-case prompts. At inference time, the model can generate test cases to self-verify and select among multiple code candidates using a pass-count criterion. On TACO and LiveCodeBench, TCS improves both pass@1 and Best-of-N performance over base and SFT baselines, with further gains when using self-generated test cases for selection. The paper releases filtered training data, model weights, and implementation details to aid reproducibility.

**Strengths:**

First, it elevates test-case generation to a first-class RL target and specifies concrete rewards for code $R_c$ and tests $R_{t1}$ and $R_{t2}$, with GRPO used for optimization and an explicit pseudocode loop. This makes the approach testable and implementable.     Second, the paper motivates a two-stage schedule by observing that $R_{t1}$ is lenient and $R_{t2}$ is stricter, then shows that $R_{t1}$ alone under the same budget gives weak scaling while adding $R_{t2}$ yields gains as $N$ grows.   Third, the inference-time selection rule is simple and auditable, defined as a pass-count objective over $N \times M$ tests, and the results indicate that self-generated test selection scales more reliably than a reward model.   Fourth, the experiments show consistent gains across two model sizes and both benchmarks, with tables and figures that break out the effect of public tests and of the two-stage reward.   Finally, the paper includes useful details for reproduction, including data filtering to a $6{,}318$-problem training set and the LiveCodeBench window, hyperparameters, and released checkpoints.

**Weaknesses:**

The Stage 1 to Stage 2 switch is described as happening after a preset test-case accuracy is reached, yet the metric, split, threshold value, and patience are not given in the main text or appendix. This limits repeatability and could affect results.

The inference-time section defines $M$ test cases per prompt, but the limitations state that the system currently produces only a single test per inference. It is not clear whether the main tables used $M=1$ or $M>1$ and how sensitive results are to $M$.

The policy-aligned buffer is a key mechanism, yet the exact buffer admission rule is given only at a high level, for example excluding timeouts and syntax errors in Stage 1 and retaining only incorrect code in Stage 2, with Algorithm 1 referring to items that satisfy the buffer policy without a precise definition or diversity constraint. A detailed admission rule and $T_b$ sensitivity would strengthen reproducibility.    For fairness across budgets, the appendix reports GPU hours and step counts yet the main results do not normalize by total RL rollouts or training tokens; a matched-budget comparison would clarify how much of the gain comes from the reward design itself.

The experimental setup uses public-case filtering in some settings and designates the first TACO evaluation case as public, which can interact with selection methods. A control without public filtering or with alternative public choices would reduce this confound.

There is at least one typo, in Algorithm 1 where “generatation” appears in the comment.

**Questions:**

Stage transition rule. You state that Stage 2 begins “upon reaching a predefined test case generation accuracy” in Stage 1. Please give the exact metric, the split used to measure it, the numerical threshold for each model size, and any patience or smoothing. If a fixed step schedule was used instead, please reconcile this with the stated accuracy trigger by citing the exact step at which you switched for each model size.

Definition and computation of $R_{t2}$. In $R_{t2}(I_g, O_g, C^{*}, C_{\text{wrong}})$ please specify how $C_{\text{wrong}}$ is chosen from the buffer $B$, how many incorrect programs are considered per update, and how you aggregate outcomes when GRPO samples a group of $G$ responses. Please also state how exceptions or timeouts during $Exec(C_{\text{wrong}}, I_g)$ affect the reward, since Stage 1 mentions excluding timeouts and syntax errors when populating $B$ but Stage 2 does not detail exception handling. A precise sampling rule and error policy are needed to reproduce $R_{t2}$.

Inference time test count $M$. Section 3.3 defines generating $M$ test cases per prompt and selecting with $N \times M$ tests using Eq. (8). The limitations section states that the current system produces only a single test case per inference. Please state the exact $M$ used in Table 1 and in the Section 4.4 runs, and if $M>1$ explain how multiple tests were produced per inference, or otherwise reconcile with the limitation.

---

> ### Author Response · Authors · 2025-12-03
>
> Thank you for your constructive comments and suggestions. Below please find our response to your questions.
>
> **Q1:** *Stage 1 to Stage 2 switch trigger.*
>
> **A1:** The stage transition is strictly based on the test case generation accuracy monitored on the training batch. Empirically, we observed that switching when the accuracy reached approximately **0.75** proved to be an effective threshold. For the 1.5B model, this threshold was reached at Step 200 (as noted in Section 4.5 and Appendix A.2); for the 7B model, applying the same 0.75 accuracy threshold corresponded to Step 40.
>
> Regarding the switching mechanism, we employed a Hard Switch strategy (transitioning at these fixed steps) without smoothing. The rationale is that the Stage 2 reward ($R_2^t$) functions as a stricter constraint imposed upon the capability learned in Stage 1 ($R_1^t$), and our experiments confirmed that a direct transition is robust and effective. To ensure deterministic reproducibility and avoid ambiguity, we fixed these step counts as the explicit triggers in our released code and experimental settings.
>
> **Revision:** We have updated Appendix A.2 to explicitly state the 0.75 accuracy threshold and its alignment with the fixed step counts for each model size.
>
> ---
>
> **Q2:** *Definition and computation of $R^t_2$.*
>
> **A2:** We clarify the sampling, aggregation, and exception handling process as follows:
>
> At each training step, we uniformly sample a batch of 128 incorrect solutions $C_{wrong}$ from the Policy-aligned Buffer. Each $C_{wrong}$ is paired with its corresponding problem description to construct a prompt using the Test Case Generation Template provided in the appendix E Prompt Template for Test Case Generation. For each prompt, we generate $G=16$ candidate test cases. Regarding the aggregation strategy, each generated test case is evaluated **only** against the specific incorrect solution $C_{wrong}$ provided in its prompt. We clarify that aggregating evaluations across multiple incorrect codes is unnecessary because the objective of our RL task is to generate a targeted counter-example for a specific error pattern. As long as the generated test case successfully exposes the flaw in the targeted $C_{wrong}$, it is considered a valid adversarial example (Reward=1). It is not required to act as a "universal" counter-example that simultaneously triggers errors in other incorrect solutions.
>
> Regarding exception handling, we distinguish between buffer admission and reward computation. While we filter out syntax errors during buffer admission to ensure basic execution capabilities, the reward computation for $R_2^t$ explicitly rewards robustness failures. Specifically, if a generated test case is valid (passed by the Oracle $C^*$) but causes the targeted $C_{wrong}$ to trigger a Runtime Error or Timeout, we consider this a successful adversarial attack and assign a reward of $R_2^t = 1$. This design encourages the model to generate corner cases that expose not only logic errors but also robustness defects (e.g., infinite loops or unhandled exceptions).
>
> **Revision:** We have incorporated the specific details regarding sampling and reward computation into a new Section A.4 of the revised manuscript.
>
> ---
>
> **Q3:** *Inference time test count $M$.*
>
> **A3:** To ensure accurate scoring of the generated test cases during training time, our current method produces only a single test case at a time, consistent with the constraint noted in the Limitations section. During the inference phase, we set $M=1$, meaning that for every generated code solution, exactly one corresponding test case is produced. Consequently, when $N$ code candidates are generated at inference time, a total of $N$ test cases are utilized for filtering the answers.
>
> **Revision:** We have added an explanation regarding the value of $M$ to Appendix A.2.
>
> ---
>
> **Q4:** *Compute Budget Fairness.*
>
> **A4:** Our experimental comparisons are strictly budget-matched to ensure fairness. As detailed in Appendix A.2, for the 1.5B model, the Code Generation Baseline is trained for 450 steps, matching the TCS model's total of 450 steps (200 steps for Stage 1 + 250 steps for Stage 2). Similarly, for the 7B model, both the baseline and the TCS model (40 steps for Stage 1 + 160 steps for Stage 2) are trained for exactly 200 steps.

---

### Official Review · Reviewer_V83i · 2025-11-01

**Soundness:** 3
**Presentation:** 2
**Contribution:** 2
**Rating:** 2
**Confidence:** 4

**Summary:**

This paper introduces Test Cases Scaling (TCS), a two-stage reinforcement learning framework designed to jointly improve a large language model's capabilities in both code generation and adversarial test case generation. The core idea is to move beyond static, predefined test suites used in typical RL-for-code setups. In Stage 1, the model is trained with RL to generate valid test cases that pass a ground-truth solution, using a reward function $R_t^1$. Once proficient, it moves to Stage 2, where it learns to generate adversarial test cases that specifically cause incorrect code solutions to fail while still passing the ground-truth solution, using a stricter reward $R_t^2$. The training process is dynamic, using a "policy-aligned buffer" of recently generated code to construct prompts for test case generation. The authors show that this method improves code generation performance (pass@1) on the TACO and LiveCodeBench datasets and, notably, enables an effective inference-time selection mechanism where the model uses its own generated test cases to select the best among multiple candidate solutions.

**Strengths:**

* The methodology is sound and the technical details are well-executed. The experiments effectively demonstrate the necessity of the two-stage approach (Figure 5) and that using self generated UT at test time is better than reward models. It also shows that the quality of the self generated test can be improved with the proposed method.
* The paper is well-written and easy to follow. The reward functions are precisely defined and their motivations are clearly explained.

**Weaknesses:**

* Reliance on Ground-Truth Solutions: The entire training framework is fundamentally dependent on the availability of a correct, executable ground-truth solution ($C^*$) to compute the rewards for test case generation (Eq. 6 and 7). This is a very strong and unrealistic assumption for a general code generation setting, where the main goal is to find solutions to problems that are not yet solved. This constraint limits the method's applicability to a form of "re-distillation" of knowledge from existing solutions, rather than enabling the discovery of new ones. Consequently, the claim that this method "extends scaling laws" is undermined, as it cannot be applied to the open-ended problems where such scaling is most needed.

*  Unclear Source of Training-Time Improvement due to Missing Baselines: The experiments do not provide a clear picture of *why* the training-time code generation performance (pass@1) improves. The comparison in Table 1 is primarily between the proposed `RL using TCS Training` and an `SFT Model`. However, two crucial baselines are missing:
    1.  **RL for Code Generation Only:** A standard RL baseline trained solely on the code generation task, using the provided ground-truth test cases as the reward signal. Without this, it is impossible to disentangle the gains from using RL in general versus the gains from the novel TCS framework. It is plausible that much of the observed improvement over SFT simply comes from the RL optimization process itself.
    2.  **Decoupled Training of Policies:** An ablation where the code generation policy and the test case generation policy are trained independently. This would test the implicit claim that joint training provides a synergistic benefit. It is unclear if the two tasks must be trained together within a single model or if one could simply train a strong, separate "verifier" model.

* The proposed adversarial framework was proposed before in other similar scenario where an external interpreter is available (e.g. for Lean with STP [1]) so the idea is not entirely novel.

[1] Dong, Kefan, and Tengyu Ma. "Stp: Self-play llm theorem provers with iterative conjecturing and proving." arXiv preprint arXiv:2502.00212 (2025).

**Questions:**

1.  The reliance on an executable ground-truth solution $C^*$ is the most significant limitation. Could you elaborate on how this method could be adapted for a more realistic setting where such solutions are not available? Have you considered alternative approaches?

2.  To better isolate the contribution of the TCS framework, could you please comment on the expected performance of the two baselines I mentioned earlier?

---

> ### Author Response · Authors · 2025-12-03
>
> Thank you for your constructive comments and suggestions. Below please find our response to your questions.
>
> **Q1:** *Reliance on Ground-Truth Solutions.*
>
> **A1:** We respectfully disagree with the notion that utilizing ground-truth solutions ($C^*$) during training limits the method to "re-distillation" or relies on unrealistic assumptions. We clarify our position through three key points:
>
> **Standard Training Paradigm vs. Inference Generalization:** Reliability on ground truth during the training phase is a standard paradigm in supervised learning and reinforcement learning with verifiable rewards [1, 2]. We use solved problems (from TACO) not to memorize solutions, but to train a robust **Verification Policy**. Crucially, **no ground truth is required at inference time.** Once trained, our model acts as an independent verifier that generates adversarial test cases to debug unseen code. Our strong performance on LiveCodeBench—a dataset of new problems collected after the training data cutoff—empirically proves that the model has learned a generalized capability to identify bugs in open-ended settings, rather than merely distilling knowledge from existing solutions.
>
> **Clarification on "Extends Scaling Laws":** The "scaling laws" mentioned in our title specifically refer to **Inference-Time Scaling** [3]. The core contribution is enabling the model to trade inference-time computation (generating $M$ test cases) for higher accuracy on unsolved problems. As shown in **Figure 4**, traditional Reward Models often struggle to scale (performance plateaus or drops due to OOD noise as samples increase). in contrast, our TCS method demonstrates consistent performance gains as the sampling budget increases. This confirms that the method effectively extends inference-time scaling to code generation tasks without needing human-in-the-loop verification.
>
> **Path to No-Ground-Truth Training:** While our current framework leverages execution feedback for maximum precision, we agree that adapting to strictly no-ground-truth training settings is a promising direction. Future work can replace the execution oracle ($C^*$) with "LLM-as-a-Verifier" approaches [4] or consistency checks [5]. However, establishing the methodology with precise execution rewards first (as done in this work) is a necessary foundation for proving the efficacy of adversarial test case generation.
>
> **Revision**: We have added relevant discussion to Appendix B Limitations of the manuscript.
>
> ---
>
> **Q2:** *Validation of Synergistic Benefits in Joint Training.*
>
> **A2:** We address the baselines requested to clarify the specific contributions of our TCS framework compared to standard RL and decoupled approaches:
>
> **Baseline 1: RL for Code Generation Only** We have already provided the "RL for Code Generation Only" baseline in **Appendix C.2**. We originally excluded it from the main text as it proved to be a weaker baseline compared to our proposed method.
>
> **Baseline 2: RL for Test Case Generation Only** For this rebuttal, we added a new baseline where the test case generation policy is trained independently ("RL for Test Case Generation Only", referred to as Model 3).
>
> The comparison results are as follows:
>
> | Model | TACO (w/o public) | LiveCodeBench (w/o public) |
> | :--- | :---: | :---: |
> | **R1-Distill-Qwen-1.5B** | 5.63 | 14.38 |
> |   + Self-Generated Test Cases | 5.78 | 22.00 |
> | **RL for Code Generation Only** | 11.16 | 18.53 |
> |   + Self-Generated Test Cases | 11.92 | 21.05 |
> | **RL for Test Case Generation Only (Model 3)** | 7.85 | 15.95 |
> | + Self-Generated Test Cases             | 15.35 | 23.26 |
> | **RL using TCS Training** | 12.31 | 20.63 |
> |   + Self-Generated Test Cases | **20.52** | **27.47** |
> | + Test Cases Generated by Model 3 | 18.79 | 25.94 |
>
> It is evident that **RL for Code Generation Only** helps code performance to some extent, but fails to generate effective test cases for filtering. In contrast, the joint method not only improves code performance but also achieves significant inference-time gains via **Self-Generated Test Cases**.
>
> While **RL for Test Case Generation Only** partially improves code generation through related knowledge, the gain is marginal compared to TCS Training. Furthermore, using test cases generated by this model to filter TCS-generated code performs significantly worse than TCS self-verification. The main reason is the **synergistic benefit** of self-play: as coding capability improves, errors become more hidden, providing higher-quality adversarial data for the test case generation objective.
>
> **Revision:** We have added **Appendix C.4** to report and analyze these experimental results.

---

> ### Author Response · Authors · 2025-12-03
>
> **Q3:** *Comparison with STP [6] and Novelty*
>
> **A3:** We agree that both TCS and STP fall under the broad umbrella of "self-play," but their mechanisms and objectives are fundamentally distinct due to the nature of their respective domains:
>
> **Problem Generation vs. Verifier Generation**:
>
> - **STP (Curriculum Learning):** STP focuses on training a "Conjecturer" to propose *new problems* (conjectures). Its goal is to create an adaptive curriculum to expose the prover to unseen theorems.
> - **TCS (Adversarial Verification):** TCS focuses on training a "Test Case Generator" to verify *existing solutions*. It does not generate new problems but instead synthesizes the *verifier itself* (test cases) to compensate for the lack of inherent formal verification in general code generation.
>
> **Role of Ground Truth:**
>
> - **STP:** Relies on external formal verifiers (Lean/Isabelle) which act as a perfect oracle to check if a proof is correct.
> - **TCS:** Operates in domains (e.g., Python) where no such perfect verifier exists. TCS *learns* to generate adversarial inputs to discriminate between correct and incorrect code, serving as a learned verifier.
>
> Therefore, despite the "self-play" label, the two methods address orthogonal challenges (Data Scarcity vs. Verification Gap) and are not interchangeable.
>
> ---
>
> [1] Shao et al. "DeepSeekMath: Pushing the Limits of Mathematical Reasoning in Open Language Models." arXiv preprint arXiv:2402.03300 (2024).
>
> [2] Le et al. "CodeRL: Mastering Code Generation through Pretrained Models and Deep Reinforcement Learning." NeurIPS (2022).
>
> [3] Wu et al. "Inference Scaling Laws: An Empirical Analysis of Compute-Optimal Inference for Problem-Solving with Language Models." arXiv preprint arXiv:2408.00724 (2024).
>
> [4] Zhang et al. "Incentivizing LLMs to Self-Verify Their Answers."  NeurIPS (2025).
>
> [5] Wang et al. "Self-Consistency Improves Chain of Thought Reasoning in Language Models." ICLR (2023).
>
> [6] Dong & Ma. "STP: Self-Play LLM Theorem Provers with Iterative Conjecturing and Proving." arXiv preprint arXiv:2502.00212 (2025).

---

### Meta-Review · Area_Chair_4Xmh · 2026-01-05

**Summary:**

1. The algorithm relies on ground truth solutions at training time (I think this concern is unfair on the authors).
2. Insufficient baselines
3. Clarity as regards stage switching (the algorithm uses two stages) and reward computation.


Note: I have looked at the fault in reward design identified by reviewer j3Nq and don't agree with the criticism.

**Reviewer Concerns:**

1. Reliance on ground truth solutions (concern unfair to authors, I disagree with the criticism).
2. Some additional experiments were provided (concerned addressed)
3. An explanation concerning state switching was provided.

**Reviewer Scores:**

V83i: 2-> 4 (main concern about training paradigm was addressed)

ChCt: 4- >6 (concern about stages was addressed)

B6xj: 4->4 (only some concerns about experiments addressed)

j3Nq: 2->2 (reviewer unlikely to change score even though their concern is largely invalid)

---

### Decision · Program_Chairs · 2026-01-26

Reject